# Anti-Tanking Pair Matching before an Elimination Phase of a Two-Phase Tournament

**Waldemar Stronka**

Department of Operations Research, Poznań University of Economics and Business, Al. Niepodległości 10, 61-875 Poznań, Poland; Waldemar.Stronka@ue.poznan.pl

**Abstract:** Perverse incentives are ubiquitous in different economic settings. In sports, they often take the form of temptation to deliberately lose matches (the phenomenon known as tanking or sandbagging). In practice, there were even such pathological situations as when a soccer team intentionally scored an own goal. We show how and when the temptation is generated by the current pair matching method, the one applied after the first phase of many popular tournaments, including the most prestigious soccer championships. If the organizers of important sporting contests do not introduce any organizational innovations, they risk serious match-fixing scandals. We introduce an alternative procedure and show that its practical implementation could radically mitigate the risk. We perform a comparative analysis of the methods. We analyze the format "Winners and Runners-up Advancing from Two Adjacent Groups", particularly its FIFA World Cup variant. In order to quantify the benefits of switching from the current method to the proposed one, we refer to simulation results. The expected decrease in temptation probability is about 83% and could be even about 90% if we additionally implement the suggested scheduling innovation.

**Keywords:** perverse incentives; tanking; sandbagging; tournament design; OR in sports

## 1. Introduction

The designers of economic systems should always analyze if the incentives they consider implementing can, under some special circumstances, act contrary to their intentions. Examples of perverse incentives are known in many different economic contexts. They are a type of the more general phenomenon of negative unintended consequences.

Perverse incentives are relatively common in sports tournaments as documented, e.g., by Kendall and Lenten (2017). They negatively affect an event's attractiveness and thus, reduce demand. Perverse incentives often lead to match-fixing, which can take the form of deliberately losing. The phenomenon (known by different names as e.g., throwing matches, tanking, or sandbagging) has been studied in the literature, for instance: Csató (2019a); Csató (2019b); Kräkel (2014).

Sports economics seems to be a promising subdiscipline for the application of the "economist as engineer" mindset advocated by Roth (2002) and others. Well-defined rules of sports make it relatively easy to analyze possible changes in the incentive structure.

We focus on incentive consequences of pair matching methods. The problem was analyzed in the literature, e.g., by Schwenk (2000) and Hwang (1982). Their practical proposals are applicable in a two-phase tournament format where after the first (group, round robin, regular season) phase, there is an elimination (knockout, playoffs) phase. Nevertheless, their ideas could be applied directly only in a single group tournament. Our paper focuses on pair matching after the group phase in a format with multiple groups. Then, there is a natural requirement that opponents in the next match have to come

from different groups. Our method proposal draws on inspiration from the suggestion in Hwang (1982, p. 238) "to let the players select their own opponents at each round but to use the official rank-order to determine which player selects first, which second and so on". If there is no contradiction in letting the group winners pick their own opponents, our method allows them to do so.

The currently applied pair matching method matches the group A winner with the group B runner-up. The extreme negative consequences of the procedure were especially visible during a soccer tournament called the Tiger Cup in 1998. The final results of group B were highly surprising. It was won by Singapore ahead of Vietnam, while Singapore was regarded as a much weaker team than Vietnam. The last group A match was between Thailand and Indonesia. Both teams were already sure to qualify to the knockout phase. For both of them, winning the match meant winning the group and, as a result, the necessity to play against strong Vietnam in the next round, while losing would lead to weaker Singapore as the next opponent. The draw was to result in Indonesia becoming the group A winner. During the game, both teams were not exerting effort in trying to win. In the last minutes of the match, when the result was a 2-2 draw, the Indonesian team captain intentionally scored an own goal despite Thai players' attempts to stop him. We emphasize that the situation would have still been undesirable, even with both team resisting the temptation of actively trying to lose. The mere occurrence of the temptation makes the match very unattractive for the fans as it is in exact opposition to the spirit of sports. They could even feel angry for the organizers for not preventing such perverse incentives.

The structure of the paper is as follows. We begin the Methodology section with explaining the analyzed tournament format. Next, we describe three pair matching methods: standard—currently applied in practice; random—representing an extreme approach to the problem of perverse incentives; and unanimity—our proposal for real world implementation. Then, we perform a theoretical analysis aimed at comparison of the standard method and the unanimity one. We are able to reach some conclusions for a general case of the considered tournament format. Then, we focus on the details of the FIFA World Cup format as a step towards quantifying the potential benefits of switching from the current method to the proposed one. The last subsection of the Methodology describes the simulation model. The Results section presents the estimates related directly to the FIFA World Cup tournament from 2018. We end with the Discussion and Conclusions.

## 2. Methodology: Comparative Analysis of Pair Matching Methods

### 2.1. Tournament Format

In practice, two-phase tournament formats consisting of a group (round robin) phase and an elimination (knockout) phase are quite popular. The key rationale for the first phase is giving all the competitors qualified for the tournament a certainty of playing at least some prespecified number of matches. Group competitors play matches with one another, gaining points. A prespecified number of competitors with the highest number of points advance to the elimination phase. Due to the tie-breaker criteria, the final group table is a strict total order (with no shared places).

The knockout phase is generally regarded as generating more climactic matches. The competitors are fully motivated not to lose, as losing means being eliminated from the tournament.

Here, we analyze a particular type of two-phase tournament format where exactly two competitors advance from every group to play further in the elimination phase. Pair matching for the first round of the elimination phase is performed separately for every pair of adjacent groups (e.g., A and B, C and D, and so on).

*2.2. Pair Matching Methods*

2.2.1. Standard

The currently applied standard pair matching method is quite simple: it matches the group winner with the adjacent group's runner-up.

The implicit assumption of the standard method is that the group winner is stronger than the runner-up. If this assumption holds, then you are best off being the group winner as it results in facing a weaker competitor advancing from the adjacent group.

The above implicit assumption is sometimes not met. Surprises are natural components of tournaments. Moreover, the organizers are not necessarily interested in minimizing the probability of surprising results. The fact that the better does not always win makes it more attractive for the fans.

If the adjacent group is won by the weaker competitor, competitors of a given group prefer to be the runner-up rather than the winner. This leads to perverse incentives if losing the match maximizes the probability of a given competitor being the runner-up rather than the winner.

2.2.2. Random

Hypothetically, it is possible to apply the following pair matching procedure, which we call the random method. There are four competitors advancing from both adjacent groups. We add the requirement that opponents in the next match have to come from different groups. The coin toss decides which of the two possible outcomes becomes the binding one.

The key feature of the random method is the fact the competitors are always indifferent between being the winner and the runner-up.

We introduce the random method just for comparison purposes. We do not believe in the applicability of the method in any real-world tournament. One of the consequences of applying the random method would be a drop in incentives to win matches. As a competitor does not gain anything by being the winner as compared to the runner-up, there are no additional incentives to become the winner. The only factor that matters is the fact of being in one of the first two places. Every time a competitor is certain to advance to the elimination phase, it loses incentives to win its remaining group matches. Obviously, matches where one or—even worse—both competitors are not motivated to win are unattractive for the fans, and consequently, not desirable for the organizers.

The random method emphasizes the threat of decreasing the desirable incentives, while trying to reduce the perverse ones.

2.2.3. Unanimity

We introduce a novel pair matching procedure called the unanimity method. This is based on preferences expressed by the group winners. The unanimity method matches the winners of two adjacent groups in one pair if both winners unanimously express their preference for playing with each other. If at least one of the winners indicates the adjacent group runner-up as its most preferred opponent, both winners play with runners-up. Thus, the outcome of the unanimity method is then identical to the standard one.

There are two basic tenets of the unanimity method. The first is that the organizers wish to reward the winners more than the runners-up. In our context, the reward is being paired with the more preferred (i.e., weaker) competitor advancing from the adjacent group. Always when rewarding both winners is possible, they both should be rewarded. The second tenet is that no one knows the competitors' preferences better than the competitors themselves. Instead of assuming something and risking a mistake, the organizers should simply ask the competitors.

The organizational side of the unanimity method could take the form of asking both winners the same question "Would you like to play the next match with the winner of the adjacent group?" and waiting for their simultaneous reply, possibly by putting their answers on the screen.

Before we present pairs matched with different methods in Table 1, we introduce the following symbols:

- $I_A$ ($I_B$)—The winner of group A (B);
- $II_A$ ($II_B$)—The runner-up of group A (B);
- $S_A$ ($S_B$)—The stronger competitor advancing from group A (B);
- $W_A$ ($W_B$)—The weaker competitor advancing from group A (B).

**Table 1.** Pairs matched with the unanimity method and the standard method.

| Case No. | First Two Places in Group Tables | Pairs Matched with the Unanimity Method | Pairs Matched with the Standard Method |
|---|---|---|---|
| 1 | $I_A = S_A$; $II_A = W_A$ <br> $I_B = S_B$; $II_B = W_B$ | $I_A$ vs. $II_B$ <br> $I_B$ vs. $II_A$ | $I_A$ vs. $II_B$ <br> $I_B$ vs. $II_A$ |
| 2 | $I_A = S_A$; $II_A = W_A$ <br> $I_B = W_B$; $II_B = S_B$ | | |
| 3 | $I_A = W_A$; $II_A = S_A$ <br> $I_B = S_B$; $II_B = W_B$ | | |
| 4 | $I_A = W_A$; $II_A = S_A$ <br> $I_B = W_B$; $II_B = S_B$ | $I_A$ vs. $I_B$ <br> $II_B$ vs. $II_A$ | |

The standard method is simply matching $I_A$ with $II_B$ and $I_B$ with $II_A$. The unanimity method is not so trivial, so we will analyze it in each of the four cases. Case 1 occurs when both adjacent groups are won by the stronger competitor. In case 1, both winners answer negatively to the question "Would you like to play the next match with the winner of the adjacent group?" so they are not paired with each other. The pairing makes them both rewarded in the sense that each winner is paired with the weaker competitor advancing from the adjacent group. In case 2, $I_A$ answers positively to the question. It would like to play with $I_B$ as the winner of group B is also the weaker competitor advancing from group B. $I_B$ answers negatively to the question because it prefers to be matched with $II_A$ as the weaker competitor advancing from group A. There is no unanimity amongst the winners, so they are not paired with each other. Rewarding both of them with a weaker opponent advancing from the adjacent group is contradictory. Case 3 is symmetrical to case 2. In case 4, both winners answer positively to the question "Would you like to play the next match with the winner of the adjacent group?" so they are paired with each other. The pairing makes them both rewarded. Case 4 is the only one where the outcomes of the standard method and the unanimity one differ. In case 4, we can see that rewarding with the standard method is the exact opposite of what would be expected. The runners-ups and not the winners are rewarded.

*2.3. Theoretical Analysis of the Incentive Impact of the Methods*

2.3.1. Format "Winners and Runners-up Advancing from Two Adjacent Groups"

In order to analyze the incentive impact of the above pair matching methods, we need to make assumptions regarding the state of affairs during the group phase. We start with assuming that:

- It is already known if group A is won by the stronger or the weaker competitor;
- The competitors advancing from group B are already known but their places are not;
- The preferences of all competitors regarding their opponents in the elimination phase are identical and this is common knowledge.

Now, we can easily make the following observation, e.g., by looking at the first two cases in Table 1. If group A is won by the stronger competitor ($I_A = S_A$), then only winning group B leads the group B competitor to be matched with the weaker competitor advancing from group A ($W_A$). The observation

holds for both methods. From the incentive perspective, it means that the two competitors already sure to advance further from group B are still motivated to win matches in their fight for the first place in the group.

The difference between the methods arises when we analyze the situation when $I_A = W_A$. Then, in the standard method, both group B competitors prefer the runner-up position to the winner one. There is a risk of temptation to deliberately lose a match. In the unanimity method, both group B competitors are indifferent between the first two places. In the unanimity method, if we know that $I_A = W_A$, we already know that the pairs are going to be: $W_A$ vs. $W_B$ and $S_A$ vs. $S_B$. From the incentive perspective, it means that the two competitors already sure to advance further from group B are not highly motivated to win but, fortunately, also not incentivized to intentionally lose.

For our further analysis, we define the "temptation to lose" as a situation when losing a match maximizes the probability of the loser being matched with the weaker competitor advancing from the adjacent group. We emphasize that it is the situation of perverse incentives which is pathological and should be reduced. For the perverse incentives to be undesirable, it is not necessary that someone actually succumbs to the temptation. The match is unattractive for the fans even if no competitor takes active efforts to lose.

To clarify the notion of "temptation to lose", we assume here that the standard method is applied and the adjacent group is won by the weaker competitor. We identify two cases in the situation when both competitors advancing from the group to the knockout phase are already known but their positions in the final group table are not. The first case is when these competitors play a match together. There is obvious "temptation to lose" for the currently first. If it loses, its probability of playing with the weaker from the adjacent group rises from zero to one. Whether there is "temptation to lose" for the currently second depends on the fact of if the match can end as a draw. If it can (as e.g., in soccer), strictly speaking, the currently second does not have "temptation to lose" but rather "temptation not to win". If the match has to end up with a decisive outcome, both competitors have "temptation to lose". The second case is when the competitors sure to advance further do not play with each other but each of them plays with a competitor already eliminated from the knockout phase. In such a case, both competitors have "temptation to lose". The competitor currently first does not know the outcome of a match played by the competitor currently second and vice versa. Nevertheless, for both of them, the loss of the match is the outcome maximizing the probability of being paired with the weaker competitor advancing from the adjacent group.

Continuing our analysis, we focus on "temptation to lose" which is faced only by competitors who are already completely sure to advance to the elimination phase. The key goal of every competitor of the group phase is advancing to the next phase. The issue of facing a stronger or weaker opponent in the first round of the elimination phase is a minor one when confronted with the threat of not advancing from the group. In general, a competitor is not going to make considerations from a pair matching perspective of potential positive consequences of losing a match if losing this match could eliminate the competitor from the tournament. Obviously, in real tournaments, there could be exceptions. For example, when there is still a theoretical possibility that a competitor will not advance to the knockout phase, but it is highly unlikely. Then, losing a match in order to get a weaker opponent could be worth taking a small risk of being eliminated from the tournament. Nevertheless, our further analysis will ignore such exceptions.

Our further analysis is performed from a single group perspective. Not taking into account the situation in the adjacent group, it is not possible to know if there is temptation in a given group. Particularly, referring to groups A and B, the temptation can never occur if it is already known that $I_A = S_A$. Then, independent of the pair matching method, the incentives in group B work exactly as we wish: as the runner-up of group B, you are never rewarded with being matched with $W_A$, independent of whether you are $S_B$ or $W_B$. To be rewarded, you have to be the winner. The necessary condition for the temptation in group B to occur is that it is certain, or at least highly likely, that $I_A = W_A$. Without additional assumptions, it is impossible to state the minimum probability required for the

necessary condition to hold. Continuing our single group analysis, we will refer to the notion of "potential temptation to lose". The "potentiality" directly refers to the adjacent group. We define the situation in a given group as "potential temptation to lose" if the above necessary condition holding in the adjacent group leads to "temptation to lose".

Now, we are ready to specify the conditions to classify a situation in a given group as "potential temptation to lose". In the standard method, the conditions are the following:

1. There is at least one competitor sure to advance to the elimination phase.
2. The group winner is still not known.

If the group winner is unknown, there are chances for the competitor already sure to qualify to be finally a runner-up. To maximize the probability of finishing in second place, the competitor should lose the remaining matches.

Contrary to the standard method where the only determining factor is the place taken by the competitor in the final group table, in the unanimity method, it is also important with whom a given competitor advances from a given group. A competitor prefers to advance with someone regarded as stronger or equivalently: everyone prefers to be the weaker. As seen in cases 3 and 4 of Table 1, to be matched with $W_A$, you have to be $W_B$ and it does not matter if you are $I_B$ or $II_B$. Being yourself certain to advance, you are incentivized to prevent qualification of someone weaker than you and help someone stronger than you to advance. If you play a match with the weaker competitor and you want to minimize the probability of your opponent's qualification, the best you can do is to win the match. The "potential temptation to lose" occurs in the unanimity method when you play with someone stronger and you can increase the probability of your opponent's qualification by losing the match.

In the unanimity method, the conditions to classify a situation in a given group as "potential temptation to lose" are the following:

1. For the competitor sure to advance to the elimination phase, it is still possible to be either the stronger or the weaker one.
2. The competitor plays a group match with an opponent stronger than itself.
3. It is still possible for this stronger opponent to qualify to the elimination phase.

The first condition implies that there is only one competitor already qualified to the knockout phase. Consequently, contrary to the standard method, at worst, only a single competitor can face the "potential temptation to lose". Moreover, it is important to notice that if it is the group favorite who is guaranteed its spot in the elimination phase, the temptation does not occur as the competitor is certain to be regarded as the stronger. This observation is important from the practical point of view as it is quite natural that most often, the strongest of all competitors in the group is the one who first ensures its qualification.

The second and third condition point to the fact that in the unanimity method, in order to classify a situation as "potential temptation to lose", you also need to know the schedule of the remaining matches. This is in contrast to the standard method, where we only need to the know the points of each competitor in the current group table.

The above conditions for both methods are general in the sense that they apply to all tournaments having the "winners and runners-up advancing from two adjacent groups" format. In order to formulate more specific conclusions, we need to refer to the particular variant of this format and take into account its details.

### 2.3.2. Format of the FIFA World Cup Type

The best-known example of the "winners and runners-up advancing from two adjacent groups" format is its variant applied in the most prestigious soccer tournament—The FIFA World Cup. The same variant is utilized in many different soccer tournaments around the world.

There are eight groups (A–H) with four teams each. Within a group, there is a round robin format. There are three rounds (match days) with two matches each. The two matches of the last round are played simultaneously. The point system for a match is 3-1-0, so 3 points for a win, 1 for a draw, and 0 for a loss. In the case of more teams having the same number of points in the final group table, the tie-breakers criteria are applied in a lexicographic way. The first tie-breaker is goal difference in all group matches.

For a competitor to be certain to qualify to the elimination phase, it needs to have a points advantage over the currently third place higher than the number of points to be gained in the remaining rounds. If the advantage is equal and not higher, there is a risk of being eliminated on the basis of the goal difference tie-breaker. It is straightforward to see that the situation of a guaranteed qualification to the knockout phase can occur no earlier than after the second round. This requires the competitor to be at least four points ahead of the currently third place.

Out of the 13 possible states of the point table after the two rounds, 4 of them indicate at least one of the competitors with sure qualification to the elimination phase. We write them in the convention: points of the currently 1st—points of the currently 2nd—points of the currently 3rd—points of the currently 4th. They are the following:

1.  6-6-0-0;
2.  6-4-1-0;
3.  6-3-1-1;
4.  6-2-1-1.

These are all the possible states of the point table satisfying the first condition to classify a situation in a given group as "potential temptation to lose" in the standard method. The last state does not satisfy the second condition as the current leader is already known to be the final group winner.

Thus, in the standard method, always if in a given group, the state of the point table after the two rounds is 6-6-0-0 or 6-4-1-0 or 6-3-1-1, there is "potential temptation to lose".

The important point we would like to make here is that there is no additional information required. As there are four teams in the group, we have 24 possible permutations. One of these teams is regarded as the strongest, one the weakest, and the others as second and third strongest. The points states specified above do not convey information about which team is in which place. This is not needed to identify "potential temptation to lose" in the standard method. Moreover, we do not need to know what matches are scheduled for the last round. If the state of the point table is one of the three specified above, the temptation occurs independent of the permutation and the schedule.

In the unanimity method, the temptation can occur only in the last three of the above states of the point table (i.e., 6-4-1-0; 6-3-1-1; 6-2-1-1). They indicate that there is exactly one competitor certain to advance to the elimination phase. However, here the states of the point table are just the necessary conditions and not the sufficient ones. In order to classify a situation as the "potential temptation to lose", we need to take into account the order of the competitors in the table as well as the matches scheduled for the third round.

To clearly identify all the situations classified as the "potential temptation to lose", we introduce the following naming convention related to the competitors of a given group. We name them by their places in the playing strength ranking. Thus, 1 is the strongest (and also, the least preferred as the opponent), 4 is the weakest (i.e., the most preferred opponent), while 3 and 4 are the second and the third strongest.

Obviously, the full schedule of the group matches is known before the group phase begins. Both the matches in the third round are played simultaneously. In our naming convention, there are the following three variants possible:

1.  Variant "1vs2" with pairs: 1 vs. 2 and 3 vs. 4;
2.  Variant "1vs3" with pairs: 1 vs. 3 and 2 vs. 4;
3.  Variant "1vs4" with pairs: 1 vs. 4 and 2 vs. 3.

In Tables 2–4, we specify situations of "potential temptation to lose" in the unanimity method.

**Table 2.** The situations of "potential temptation to lose" in the unanimity method when the last round matches are 1 vs. 2 and 3 vs. 4.

| States of the Point Table | Order of the Competitors |
|:---:|:---:|
| 6-4-1-0 | 2-1-3-4 |
| | 2-1-4-3 |
| 6-3-1-1 | 2-3-1-4 |
| | 2-4-1-3 |
| | 2-3-4-1 |
| | 2-4-3-1 |
| 6-2-1-1 | 2-1-3-4 |
| | 2-1-4-3 |

**Table 3.** The situations of "potential temptation to lose" in the unanimity method when the last round matches are 1 vs. 3 and 2 vs. 4.

| States of the Point Table | Order of the Competitors |
|:---:|:---:|
| 6-4-1-0 | 3-1-2-4 |
| | 3-1-4-2 |
| 6-3-1-1 | 3-2-1-4 |
| | 3-4-1-2 |
| | 3-2-4-1 |
| | 3-4-2-1 |
| 6-2-1-1 | 3-1-2-4 |
| | 3-1-4-2 |

**Table 4.** The situations of "potential temptation to lose" in the unanimity method when the last round matches are 1 vs. 4 and 2 vs. 3.

| States of the Point Table | Order of the Competitors |
|:---:|:---:|
| 6-4-1-0 | 3-2-1-4 |
| | 3-2-4-1 |
| 6-3-1-1 | 3-1-2-4 |
| | 3-4-2-1 |
| | 3-1-4-2 |
| | 3-4-1-2 |
| 6-2-1-1 | 3-2-1-4 |
| | 3-2-4-1 |

The very nature of the process of gaining points by competitors implies which current places can meet in the third round. If the state of the point table is 6-4-1-0 or 6-2-1-1, the competitors from the two currently highest places play against each other in the last round. If the state of the point table is 6-2-1-1, the current leader plays the last match either with the currently third place or the fourth one.

*2.4. Simulation Model*

We refer directly to the FIFA World Cup 2018 and perform the calculations for each group separately. Our calculations are based on the model and numerical values proposed in Dormagen (2014) applying the Elo rating points from the World Elo Ratings website (eloratings.net). We took the latest available data before the event—from 31 December 2017.

To calculate the probabilities of three possible outcomes (favorite win; draw; favorite loss) of a given match, we start with calculating the win expectancy of the favorite ($W_{Fav}$). The formula is the following:

$$W_{Fav} = \frac{1}{1 + 10^{-\frac{r_1 - r_2}{400}}} \tag{1}$$

where:

$r_1$—The Elo rating points of the higher rated opponent in the match (i.e., the match favorite);

$r_2$—The Elo rating points of the lower rated opponent in the match.

Next, we calculate the probability of the match being drawn:

$$P_{Draw} = \frac{1}{3} \cdot \exp\left(-\frac{(W_{Fav} - 0.5)^2}{2 \times 0.28^2}\right) \tag{2}$$

Then, it is straightforward to calculate the probability of the favorite winning the match:

$$P_{Fav\_Win} = W_{Fav} \cdot (1 - P_{Draw}) \tag{3}$$

Finally, obviously:

$$P_{Fav\_Loss} = 1 - P_{Fav\_Win} - P_{Draw} \tag{4}$$

With the model for calculating probabilities of outcomes in all matches, we can proceed to calculate the probabilities of "potential temptation to lose". We are interested in the situation after the first two rounds, so after the four group matches. Each match has three possible outcomes (favorite win; draw; favorite loss), so we have to consider $3^4 = 81$ scenarios. The probability of the scenario is the product of four match outcome probabilities.

In each group, we simulate the 81 scenarios for three different variants of the last round matches ("1vs2", "1vs3", "1vs4"). As there are eight groups at the FIFA World Cup, we have 24 sets of 81 scenarios. The calculations for each set are placed in a different sheet of our spreadsheet.

For both methods, each of the 81 scenarios is classified either as the "potential temptation to lose" situation or not. Five of them are situations of "potential temptation to lose" in the unanimity method and 18 of them in the standard method. Out of the five situations of "potential temptation to lose" in the unanimity method, four are in common with the standard one. The only exception is related to 6-2-1-1 state of the point table. The sum of 18 (5) probabilities is the probability of the "potential temptation to lose" for the standard (unanimity) method in a given group with a given last round variant.

Now, we can estimate the probability of "temptation to lose" in a single pair of two adjacent groups. The vast majority of "temptation to lose" cases occur in last round matches of the group finishing its matches later than its adjacent group. In the FIFA World Cup, groups A, C, E, and G (called: "earlier groups") finish matches one day earlier than groups B, D, F, and H (called: "later groups"). However, to be precise, we have to mention that there is a possibility of "temptation to lose" occurring in the earlier group. It can occur when after two rounds, the state of the point table in the later group is 6-2-1-1 and its leader is the weakest team (i.e., 4). Then, the earlier group competitors know for sure before their last matches that the adjacent group will be won by the weaker team than the runner-up. Such situations have very low probabilities and we do not take them into account in the simulations.

In our simulations, we assume that the "temptation to lose" in a pair of adjacent groups occurs when the earlier group is won by the weaker team than the runner-up and there is a "potential temptation to lose" in the later group. As we apply the Elo-based model for calculating match outcome probabilities, we assume that both events are independent. We are interested in estimating the probability of "temptation to lose" in the FIFA World Cup tournaments in general, not particularly in 2018. The fact of which groups were earlier and which were later in the 2018 tournament is not

important for our general estimation. We could equally well have any different "shuffling" of the letters between the groups, including the one where the earlier and later groups are reversed. Thus, we assume that the probability of "temptation to lose" in a random pair of adjacent groups is a product of the probability of the "potential temptation to lose" in a random group and the probability of a random group being won by the weaker team than the runner-up.

To calculate the probability of a random group being won by the weaker team than the runner-up, we generated 729 ($3^6$) scenarios for each group. In the case of more teams having the same number of points in the final group table, we implemented the random tie-breaker. Technically, for each of the 729 scenarios, we generated 24 additional tie-breaker scenarios with equal probability of occurrence. These additional scenarios are all possible permutations of the fractional points (0.1; 0.2; 0.3; 0.4). The results are presented in our spreadsheet.

## 3. Results

The results of the simulations are summarized in Tables 5–8.

**Table 5.** Probabilities of the "potential temptation to lose" in the standard method.

| Variant of the Last Round | Standard | | | | | | | | Average |
| :---: | :---: | :---: | :---: | :---: | :---: | :---: | :---: | :---: | :---: |
| | Group | | | | | | | | |
| | A | B | C | D | E | F | G | H | |
| 1vs2 | 0.35 | 0.48 | 0.32 | 0.34 | 0.44 | 0.38 | 0.52 | 0.33 | **0.40** |
| 1vs3 | 0.31 | 0.24 | 0.28 | 0.29 | 0.33 | 0.36 | 0.24 | 0.25 | **0.29** |
| 1vs4 | 0.26 | 0.23 | 0.24 | 0.24 | 0.32 | 0.31 | 0.24 | 0.24 | **0.26** |
| Average | **0.31** | **0.32** | **0.28** | **0.29** | **0.36** | **0.35** | **0.33** | **0.28** | 0.31 |

**Table 6.** Probabilities of the "potential temptation to lose" in the unanimity method.

| Variant of the Last Round | Unanimity | | | | | | | | Average |
| :---: | :---: | :---: | :---: | :---: | :---: | :---: | :---: | :---: | :---: |
| | Group | | | | | | | | |
| | A | B | C | D | E | F | G | H | |
| 1vs2 | 0.06 | 0.13 | 0.08 | 0.07 | 0.05 | 0.04 | 0.14 | 0.09 | **0.08** |
| 1vs3 | 0.05 | 0.04 | 0.06 | 0.05 | 0.03 | 0.04 | 0.03 | 0.04 | **0.04** |
| 1vs4 | 0.04 | 0.03 | 0.05 | 0.04 | 0.02 | 0.02 | 0.03 | 0.04 | **0.03** |
| Average | **0.05** | **0.07** | **0.06** | **0.06** | **0.03** | **0.03** | **0.06** | **0.06** | 0.05 |

**Table 7.** Ratios of the "potential temptation to lose" probabilities for the analyzed methods.

| Variant of the Last Round | Ratios: Standard/Unanimity | | | | | | | | Average |
| :---: | :---: | :---: | :---: | :---: | :---: | :---: | :---: | :---: | :---: |
| | Group | | | | | | | | |
| | A | B | C | D | E | F | G | H | |
| 1vs2 | 5.7 | 3.7 | 4.2 | 4.6 | 8.4 | 9.0 | 3.6 | 3.8 | **4.8** |
| 1vs3 | 6.6 | 6.5 | 4.8 | 5.4 | 12.7 | 9.9 | 9.3 | 5.8 | **7.0** |
| 1vs4 | 7.2 | 6.9 | 4.8 | 5.5 | 20.4 | 13.0 | 9.4 | 6.3 | **7.8** |
| Average | **6.4** | **4.8** | **4.5** | **5.1** | **11.6** | **10.2** | **5.1** | **4.9** | 6.0 |

The key result is that the probability of potential temptation to lose for the standard method is about six times higher than for the unanimity one. If we switch from the standard to the unanimity method, we could expect the probability of the "potential temptation to lose" in a random group to be reduced from 0.31 to 0.05. It clearly shows that implementing the proposed method would not be a marginal improvement but rather a radical change.

We see that even the smallest improvements as observed over all groups and last round variants lead to a reduction in probability by a factor of more than 3. Moreover, the highest value for the unanimity method (0.14) is still much lower than the lowest probability for the standard method (0.23).

The interesting observation is that for both methods and for each group, the 1 vs. 4 variant of the last round leads to the lowest probabilities, while the 1 vs. 2 variant to the highest. Moreover, the 1 vs. 4 variant is also the one when there is the highest relative benefit from applying the unanimity method instead of the standard one. Currently, each of the three variants seems to have a similar frequency of occurrence in practice. However, our analysis shows that the variant 1 vs. 4 should be preferred. If, together with switching to the unanimity method, organizers decided also to always choose the 1 vs. 4 variant, they could expect the decrease in the probability of "potential temptation to lose" by one order of magnitude (from 0.31 to about 0.03).

From Table 8, we see that the probability of a random group being won by the weaker team than the runner-up is about 0.34. Thus, we could conclude that while currently the probability of the "temptation to lose" in a random pair of adjacent groups is about 0.1 ($\approx$0.31 $\cdot$ 0.34), switching to the unanimity method and implementing the scheduling innovation can decrease it to about 0.01 ($\approx$0.03 $\cdot$ 0.34).

**Table 8.** Probabilities of the group being won by the weaker team than the runner-up.

| Group | | | | | | | | Average |
|---|---|---|---|---|---|---|---|---|
| **A** | **B** | **C** | **D** | **E** | **F** | **G** | **H** | |
| 0.32 | 0.39 | 0.38 | 0.35 | 0.23 | 0.24 | 0.41 | 0.38 | **0.34** |

## 4. Discussion

In our analysis, we focused only on the temptation to lose, which is the result of a desire to meet the weaker opponent in the first round of the knockout phase. In practice, such temptation could also have different causes, see e.g., Caruso (2009); Andreff (2019). One of them is simply bribery. Nevertheless, there are also more subtle causes related to the tournament organization. For example, intentional loss is sometimes a way to eliminate a strong (inconvenient) rival from the later part of the tournament. Taking such a possibility into account, it is no longer true that when the standard (or random) method is applied, competitors are indifferent to who advances with them from the same group. In the standard method, the temptation to lose aimed at eliminating someone pose a risk particularly when the group winner is already known. As the known winner has nothing to lose and nothing to gain in terms of the next round, it can be tempted to think in the longer run. This seems to suggest that we should construct the utility function of the competitors not only dependent on the chances in the first round of the elimination phase, but also taking into account the later ones. However, we could still think of different reasonable factors which should also be reflected in the utility function; for example, the fact that some competitors in the last group matches are highly interested in saving energy for the next match or minimizing the risk of injuries. Again, this is particularly clear for the sure group winners. Generally, the analysis with more realistic competitors' utilities would certainly be an interesting continuation of the work presented in this paper. Nevertheless, we should be aware that, as the task of reflecting the competitors' utilities is complex, it will be difficult to formulate a relatively unquestionable set of assumptions.

Certainly, we could ask if in our simulations, the assumptions on calculating match outcome probabilities from the Elo rating points are not too simplified. For example, we could claim that they are not constant in time as they depend on the outcomes of the previous matches. There are many approaches to modeling outcomes of sports matches. In the paper, our key goal was more qualitative than quantitative in nature. We wanted to know if the benefit of switching into the unanimity method is more of a radical change or just a small improvement. Performing the simulations with different

outcome modeling approaches would certainly be interesting. However, it is difficult to see how they could lead to qualitatively different conclusions.

Although, the unanimity method does not exclude the possibility of temptation to lose in general, we claim that it completely eliminates the most extreme cases of such temptation. The first case occurs when in both last group matches, there is an opponent experiencing the temptation. Such double temptations are impossible in the unanimity method. If the proposed method is applied, there can be, at most, one competitor in the group facing the temptation. The second extreme case is when no one wants to win a match. In sports not allowing draws as the final match outcomes, both competitors can be tempted to lose. With a draw accepted as the final outcome (as e.g., in soccer), there is a match between an opponent with "temptation to lose" with the one having the "temptation not to win". Such a possibility is also excluded in the unanimity method where if the competitor faces "temptation to lose", its opponent always wants to win. Matches of the above kind can look the most pathological, including the events similar to the ones which, as we mentioned earlier, happened during the Tiger Cup 1998. If one of the opponents aims at winning, for the side experiencing "temptation to lose", it could often be enough to slightly reduce effort and energy exerted in play in order to really lose. For non-expert viewers, such a match can look almost typical. The "acting" performed by the tempted opponent does not necessarily have to be clearly visible in order to be effective. When both sides do not actively try to win, viewers of the match could identify the contradiction of the sporting spirit much more easily. Thus, such matches can be especially harmful for the image of the tournament and its organizers.

## 5. Conclusions

The new pair matching procedure—the unanimity method—introduced in the paper does not aim at completely eliminating the temptation to lose. As clearly shown with the example of the random method, the complete elimination of temptations could lead to serious side effects in the form of a significant reduction in desirable incentives to win. The unanimity method does not cause such side effects while achieving a large decrease in the temptations.

Our simulation results suggest a drop in the probability of temptation by about 83%. Naturally, the exact number is highly dependent on the detailed model assumptions. From the practical perspective, the key is that the decrease resulting from switching from the current standard method to the proposed unanimity one is quite large. It is not the case that only a small marginal improvement is achievable and, in order to achieve it, we have to risk implementing a new procedure completely unknown to the fans accustomed to the current practice.

Somehow unexpectedly, our simulations show that changing the pair matching method is not the only way the organizers can minimize the temptation to lose. It turned out that the group match schedule also plays a role. In particular, the schedule minimizing temptation is the one where the strongest and the weakest competitors in the group meet in the third round. Thus, instead of not favoring any variant of last round matches, organizers should always schedule the 1 vs. 4 match in the last round. Adding this schedule innovation to the change in the pair matching method would increase the drop in the probability of temptation from about 83% to about 90%.

**Funding:** This research received no external funding.

**Conflicts of Interest:** The author declares no conflict of interest.

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
