# Peer review of "Anti-Tanking Pair Matching before an Elimination Phase of a Two-Phase Tournament"

_economies, doi:10.3390/economies8030066_

Round 1
Reviewer 1 Report
This paper points out a relatively minor flaw in the design of football tournaments, such as the World Cup and Euro tournaments, and then suggests an improvement to the design. In these tournaments, typically 4 teams are assigned to each group that plays a round-robin tournament, each playing each other once. The winner of each group (as determined by points) plays the second place team from another group whose group is determined by FIFA rules in advance. If the winner of any one group is an inferior team to the second place finisher, perhaps due to an unexpected surge in performance, injuries, etc., there is an incentive for top talented team in another group to finish in second place in their group so as to play the inferior first place team in the other group. This could result in the top team in Group A losing its last game on purpose to an inferior team in order to be seeded against the first place team in Group B that is an inferior team. Although such strategic behavior is a possibility, it is not clear how often such a situation emerges in a top football tournament. Two teams of equal ability to win Group A could each choose to lose their final matches, each playing an inferior team in Group A. This is why FIFA requires that the final matches in each group be played simultaneously to prevent collusion between teams.
The author(s) suggest that the winner of each group be allowed to select whether to play the first or second place team in the opposing group for the knockout portion of the tournament. This simple fix addresses the problem and the author(s) demonstrate through some simulated winning probabilities how incentives will change. However I would think that when winning the group is rewarded with the choice of which team to play, teams might try harder to win the group instead of finishing second in most cases. It is only the case where an inferior team wins Group B that the best team in Group A benefits by finishing second. I found the many permutations of possible outcomes in the middle section of the paper hard to work through. Perhaps this could be broken down into a few short sections using some real teams, instead of the rather clumsy notation.
The timing of the group stage matches is critical to the author(s) results. For instance, suppose all final Group A matches are played on Monday and all final Group B matches on Tuesday. The top two teams in Group A then do not know the final order of finish in Group B when playing their final matches on Monday. The top 2 teams in Group B play on Tuesday and will know the final two places in Group A, so they gain an advantage. The ability to choose the knockout round opponent does not aid the top 2 teams in Group A. The author(s) are implicitly assuming that all teams know the order of finish of their opposing group, but that can only be true for one-half of the groups. I am not sure if the author(s) considered this and it will change the results in their simulations for sure, perhaps so much so, that it renders their results questionable. Any simulation will have to account for the timing of the matches in the groups.
Reviewer 2 Report
See attached

Round 2
Reviewer 1 Report
The author(s) have addressed my initial concerns. It would be useful to mention how the probability of a draw is computed. I am assuming it is the usual method used by bettors where P(Draw) = 0.30 - 0.01(r1-r2-100) where r1 is the higher ELO rating. If r1-r2 < 100 then P(Draw) = 0.30. This is rather important to the simulations.
Author Response
The explicit formula for P(draw) is given in section 2.4.
The reference is also given: (Dormagen 2014), link: http://www.mi.fu-berlin.de/inf/groups/ag-ki/Theses/Completed-theses/Bachelor-theses/2014/Dormagen/Bachelor-Dormagen.pdf
The details of P(draw) calculation are described in section 3.3. (p. 17) of this paper.